# Modification and Application of Bamboo-Based Materials: A Review—Part II: Application of Bamboo-Based Materials

Zhichao Lou [1,*,†], Zhiyu Zheng [2,3,†], Nina Yan [2,3], Xizhi Jiang [2,3], Xiaomei Zhang [2,3], Shan Chen [2,3], Rui Xu [2,3], Chun Liu [2,3] and Lei Xu [2,3,*]

[1] Jiangsu Co-Innovation Center of Efficient Processing and Utilization of Forest Resources, Nanjing Forestry University, Nanjing 210037, China

[2] Jiangsu Engineering Technology Research Center of Biomass Composites and Addictive Manufacturing, Institute of Agricultural Facilities and Equipment, Jiangsu Academy of Agricultural Sciences, Nanjing 210014, China; zhiyu7875@gmail.com (Z.Z.); yannina@jaas.ac.cn (N.Y.); jiangxizhi@jaas.ac.cn (X.J.); zhangxiaomei@jaas.ac.cn (X.Z.); chenshan@jaas.ac.cn (S.C.); 20210080@jaas.ac.cn (R.X.); 20180075@jaas.ac.cn (C.L.)

[3] Key Laboratory for Protected Agricultural Engineering in the Middle and Lower Reaches of Yangtze River, Ministry of Agriculture and Rural Affairs, Nanjing 210014, China

* Correspondence: zc-lou2015@njfu.edu.cn (Z.L.); xulei@jaas.ac.cn (L.X.)
† These authors contributed equally to this work.

**Abstract:** Bamboo, with its inherently porous composition and exceptional renewability, stands as a symbolic embodiment of sustainability. The imperative to fortify the utilization of bamboo-based materials becomes paramount for future developments. These materials not only find direct applications in the construction and furniture sectors but also exhibit versatility in burgeoning domains such as adsorption materials and electrode components, thereby expanding their consequential influence. This comprehensive review meticulously delves into both their explicit applications and the nuanced panorama of derived uses, thereby illuminating the multifaceted nature of bamboo-based materials. Beyond their current roles, these materials hold promise for addressing environmental challenges and serving as eco-friendly alternatives across diverse industries. Lastly, we provide some insights into the future prospects of bamboo-based materials, which are poised to lead the way in further development. In conclusion, bamboo-based materials hold immense potential across diverse domains and are set to play an increasingly pivotal role in sustainable development.

**Keywords:** bamboo; sustainability; application

## 1. Introduction

With its rapid growth, bamboo is often referred to as the "green gold" of the plant kingdom and is a symbol of low-carbon, sustainable development. Bamboo is renowned for its strength and durability, despite its lightweight nature. Its fibrous structure provides impressive tensile strength, making it ideal for uses in fabrics [1] and furniture [2,3] and even as a building material [4–6]. For example, Yang et al. utilized sodium percarbonate, an environmentally friendly oxidant, in conjunction with the alkali oxygen bath technique to extract fibers from bamboo shoot shells. This material is viewed as a viable and sustainable resource for the textile industry [7]. Shi et al. discovered that bamboo self-bonded composites, made using bamboo powder with moisture content of 40%, exhibited significant enhancements in strength and excellent adhesion for paint films [3]. These findings establish it as an exceptional material for furniture production. For building materials, Li et al. developed a bamboo scrimber composite material with a rupture modulus of 196.6 MPa and an elastic modulus of 18.5 GPa using bamboo fiber mats and phenolic resin as raw materials; this material is a promising substitute for traditional building bamboo products [8]. Bamboo's remarkable versatility also extends to various aspects of daily

life, encompassing applications in paper production, bamboo toothpick manufacturing, reusable utensils, and much more [9].

In addition to its practical applications, bamboo plays a crucial role in the chemical industry, including pollutant treatment [10], serving as an electrode material [11], contributing to the generation of bioenergy [12], etc. This is attributed to the exceptional properties of bamboo, which impart distinct characteristics to its derivatives, thereby broadening their range of applications. For example, bamboo-based biochar, owing to its exceptional pore structure, is not only an exceptional adsorbent but also exhibits significant potential as an electrode material for supercapacitors [13–16]. Its unique pore structure provides abundant adsorption sites, making it highly effective in areas such as pollutant removal and water purification. Simultaneously, these pores offer an ideal space for charge storage, making bamboo-based biochar a subject of interest in energy storage, especially as an electrode material for supercapacitors [17]. Its electrochemical performance and excellent conductivity play a crucial role in high-performance energy storage devices, offering a potential solution for renewable energy storage.

In general, bamboo-based materials, as green and sustainable materials, hold significant potential in various future application areas, making them indispensable green materials. However, in the present review, only research within a specific domain of bamboo is discussed, and there is a limited comprehensive overview of both the direct and derived applications of bamboo. In this review, we not only introduce the direct applications of bamboo, such as construction, furniture, etc., but also introduce the applications of bamboo-derived materials, such as adsorption, electrode materials, etc. Lastly, this article addresses some of the challenges and provides future prospects for novel bamboo-based materials. This comprehensive review contributes by consolidating knowledge and presenting it in a structured and accessible format, serving as a valuable resource for anyone interested in bamboo applications and their potential impact across diverse sectors. It is our hope that this review will enhance understanding of bamboo-based materials and inspire new research directions to further advance this field. The sustainability and versatility of bamboo materials make them powerful tools for addressing contemporary global challenges and are poised to continue playing a pivotal role in various application areas. Figure 1 shows the structural characteristics and current applications of bamboo. The abbreviations covered in this article are listed in Abbreviations section.

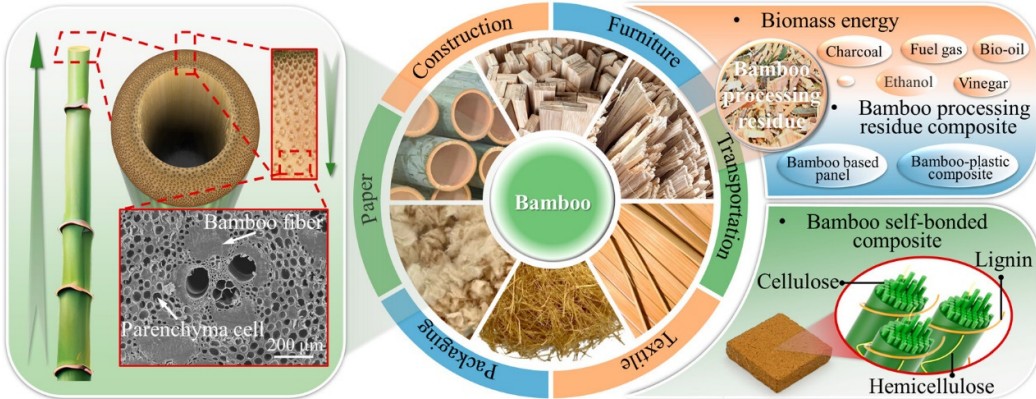

**Figure 1.** Structural characteristics and current applications of bamboo [3].

## 2. Direct Application of Bamboo-Based Materials

### 2.1. Building Materials

Bamboo-based building materials are processed from natural bamboo into building materials with various dimensions using advanced composite and recombination technology. They are ideal green materials for the construction industry [18]. Modern building bamboo materials mainly include raw bamboo and engineered bamboo (glued bamboo, reconstituted bamboo, composite bamboo).

### 2.1.1. Raw Bamboo Building Materials

Raw bamboo refers to bamboo that retains the original character of round bamboo and bamboo slices. As a natural material with obvious initial defects, it is necessary to study the strength grading and design the connection properties of bamboo [19,20]. Clamp moment connections offer a viable alternative for enhancing the structural performance and versatility of bamboo structures. Monrad et al. introduced three novel beam–column bamboo connections designed to effectively transmit moments. The stiffness, strength, and ductility of the three designs were in the ranges of 73.4–230.8 kN·m/rad, 3.4–5.2 kN·m, and 3.6–17.2, respectively. In comparison to traditional mortar-injected bolted connections, the average stiffness and strength of these connections demonstrated notable increases, with minimum improvements of 29% in stiffness and 250% in strength [21].

### 2.1.2. Engineered Bamboo Building Materials

With the further study of raw bamboo, a variety of structural engineered bamboo products have appeared in recent years, such as glued bamboo (glubam), reconstituted bamboo, composite bamboo, and other new bamboo materials.

Glued bamboo represents a natural composite material reinforced with bamboo fibers that has specific fiber arrangement and special processing [22]. The assembly method and treatment processing affect the bonding performance of glued bamboo. Meng et al. reported that an in situ glued-bamboo (glubam) composite demonstrated record-high shear strength of approximately 4.4 MPa and tensile strength of around 300 MPa [23]. The experimental flow chart and comparison of results are shown in Figure 2. The shear strength of laminated bamboo lumber was increased after the ultrasonic treatment of bamboo strips [24].

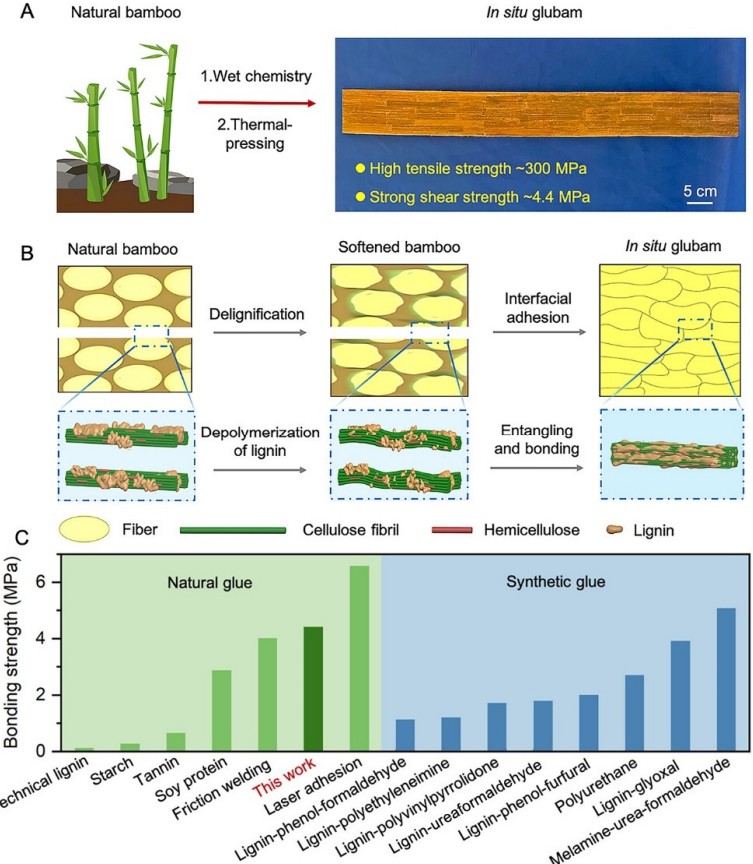

**Figure 2.** (**A**) Natural bamboo is processed into scalable in situ glubam using alkaline and thermal-pressing treatments. (**B**) Schematic illustrations of the transformation of natural bamboo into in situ glubam. (**C**) Bonding strength of in situ lignin glue compared with that of other natural and synthetic glues [23].

Reconstituted bamboo is a kind of green and high-strength bamboo-based material made from raw bamboo substrates. It is produced through various processes, such as strip cutting, low-temperature drying, hot-pressing gluing. Many researchers have studied the mechanical properties of reconstituted bamboo, providing performance indicators, failure modes, and constitutive models for different forms of reconstituted bamboo [25–28].

To further enhance the properties of bamboo, many new composite materials have been proposed by combining bamboo with other materials. Among them, bamboo–timber, bamboo–concrete, and bamboo–steel composites are common forms. Bamboo–timber composite material is a board made from wood and bamboo as the main raw materials and is laminated and glued along the grain direction using the method of laminated wood veneer. Li et al. examined the post-yielding behavior of five-layer cross-laminated bamboo and timber short-wall specimens under in-plane compressive loading, with cross-laminated timber specimens serving as the control. The incorporation of bamboo scrimber layers resulted in a remarkable 90% improvement in compressive strength and a 20% increase in compressive modulus in the major strength direction [29]. Cross-laminated bamboo and timber beams have better bending stiffness and capacity than cross-laminated timber beams under in-plane bending conditions [30]. Shan et al. documented the shear–slip behaviors of different connections used in composite beams consisting of a concrete slab and glue-laminated bamboo (glubam) beam [31]. Fiber-reinforced polymer (FRP), concrete, and bamboo were combined to form FRP–bamboo–concrete composite beams with higher ultimate load and cross-section stiffness [32,33]. The ultimate bearing capacity, flexural rigidity, ductility, and material utilization efficiency of bamboo beams can be effectively improved by embedding steel bars and pre-stressed steel bars, because a stiffened bamboo beam can produce reverse-bending deformation under the action of service load and reduce the actual deflection [34,35].

### 2.1.3. Functional Bamboo Building Materials

The functional utilization of bamboo-based materials in architecture is also the focus of research. Huang et al. utilized natural bamboo fiber and bamboo charcoal as local construction infills in the building envelope. The strategic placement of bamboo charcoal on the upstream side of the moisture flow enhanced the hygric performance of the bamboo fiber layer, mitigated moisture and heat flow through exterior walls, and improved the indoor hygrothermal environment [36]. On the other hand, Li et al. extracted sustainable high-strength macrofibers from natural bamboo, which exhibited impressive mechanical properties, including tensile strength of $1.90 \pm 0.32$ GPa, Young's modulus of $91.3 \pm 29.7$ GPa, toughness of $25.4 \pm 4.5$ MJ m$^{-3}$, and specific strength of $1.26 \pm 0.21$ GPa cm$^{-3}$ g$^{-1}$ [37].

Bamboo-based materials exhibit enhanced mechanical properties, dense microscopic characteristics, and shorter sustainable supply cycles, which can be further improved with physical or chemical methods. They represent a viable natural, high-quality alternative for construction applications.

### 2.2. Furniture Materials

Bamboo furniture is both natural and eco-friendly, making it a novel option for modern homes and positioning bamboo as an exceptional raw material for the development of high-quality biocomposites. However, using bamboo directly can lead to certain issues, such as cracking and inadequate strength. To address the expansion and weakening issues observed in bamboo utilized for structural and furniture materials [38–40], extensive research has been conducted to enhance the properties of bamboo materials in furniture applications.

For example, Ge et al. introduced an innovative approach to transform bamboo biomass into a natural, sustainable, fiber-based biocomposite for structural and furniture applications. This method involves hot pressing and low energy consumption and excludes the use of adhesives [41]. The biocomposite incorporates a significant 10-fold improvement in internal bonding strength, along with enhanced water resistance, improved fire safety,

and environmentally friendly properties. This advancement is in contrast to current furniture materials that rely on hazardous formaldehyde-based adhesives. The enhanced fire and water resistance of the biocomposite eliminate the need for toxic adhesives, which are commonly derived from formaldehyde-based resin. This alleviates concerns about harmful formaldehyde-based volatile organic compound (VOC) emissions, ensuring improved indoor air quality. A noteworthy aspect of this approach lies in its ability to fully convert discarded bamboo biomass into biocomposite, offering a potentially cost-effective alternative with high revenue potential. Shi et al. conducted an investigation into the influence of process parameters (bamboo powder moisture content, hot-pressing temperature, and target density) on the microstructure, physical properties, and mechanical performance of bamboo self-bonded composites (BSCs) intended for furniture applications. The findings revealed that BSCs prepared from bamboo powder with a moisture content of 40% exhibited the best water resistance and mechanical properties. Higher temperatures and target densities were found to facilitate the formation of denser structures, significantly enhancing water resistance and mechanical performance. This combination yielded high-performance BSCs with a thickness expansion rate of 12.8% and internal bonding strength of 0.71 MPa, making them comparable to commercially available, furniture-grade, medium-density fiberboards. Moreover, these BSCs exhibited excellent paint adhesion and hold substantial potential for furniture applications [3].

In order to explore the application of bamboo in furniture, Fu et al. used ANSYS finite element analysis software to compare the stress and deformation of reconstituted bamboo under loading [42]. This introduces a novel perspective for advancing modern furniture design methodologies. Yuan et al. [2] illustrate four bamboo flattening technologies, as depicted in Figure 3. These technologies facilitate the efficient and value-added utilization of bamboo in the production of furniture and engineered composites. Through the analysis of existing furniture cases and the coordination evolution rule of the TRIZ system, Dong et al. [43] aid designers in gaining a comprehensive understanding of existing technology. In order to assess the feasibility of employing environmentally friendly composite materials for street furniture, Vengala et al. conducted pertinent research, ensuring greater stability of bamboo reinforcements and further enhancing the durability of furniture benches [44]. Concrete furniture elements can be manufactured using bamboo as reinforcement, ensuring the stability of the reinforcement without requiring vibration for concrete consolidation. The performance study conducted on the bench demonstrated its ability to withstand a load significantly greater than the design load, with minimal deflection. The bench's load-carrying capacity was nearly two and a half times the design load. A cost analysis indicated that the precast concrete bench prepared with bamboo reinforcement was 18% more cost-effective than a bench made with steel reinforcement. Suhaily et al. analyzed the physical, mechanical, and morphological properties of a five-layer composite material composed of dry bamboo strips and oil palm trunk veneer in order to explore its feasibility as an alternative material for furniture manufacturing [45]. The composite exhibited the highest density when arranged perpendicularly, and it displayed lower water absorption behavior. Perpendicular-arrangement composites showed almost similar patterns in thickness swelling. Regarding the mechanical properties, the layer arrangements significantly influenced the mechanical properties of both composites. In the majority of cases, arrangements in a perpendicular orientation demonstrated higher strength compared with parallel arrangements in both hardness and impact strength tests. The morphology of the composite that was composed of dry bamboo strips and oil palm trunk veneer verified the high content of silica in oil palm trunk veneer. Prototype testing confirmed that the table design and materials used met stability requirements. The arrangement of layers in a perpendicular orientation helped maintain higher strength during impact tests for the table. This research suggests that sustained efforts and focused attention can promote the utilization of bamboo hybrid composites as alternative raw materials to wood. This, in turn, encourages the adoption of sustainable lifestyles and innovative furniture design among users.

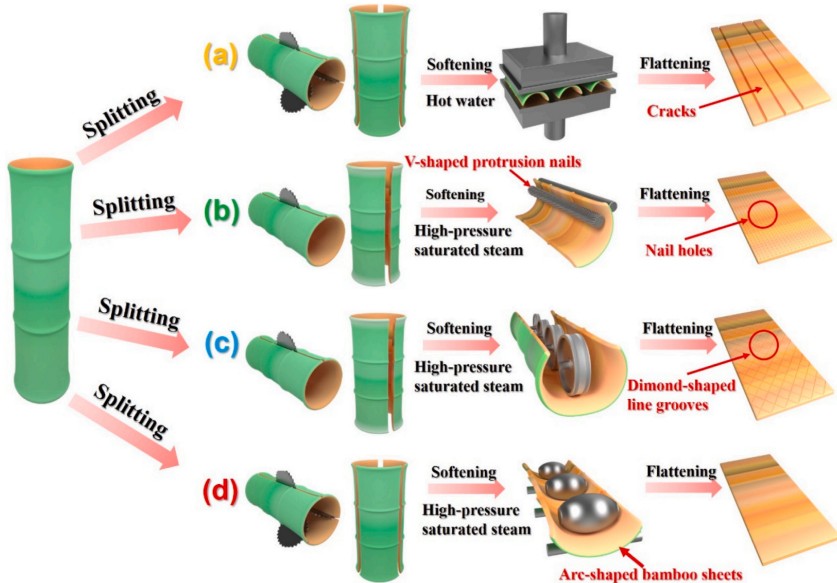

**Figure 3.** Four types of bamboo flattening technology: (**a**) one-step pressure flattening; (**b**,**c**): two types of notched-flattening technologies; (**d**) non-notched-flattening technology [2].

Promoting recombinant bamboo furniture proves to be an effective solution for addressing the shortage of wood [46]. Recombinant bamboo not only surpasses ordinary timber in mechanical strength and features a beautiful grain color but also exhibits processability comparable to hardwood [47]. As a result, it stands as an ideal material for furniture manufacturing.

### 2.3. Biofuel/Energy

#### 2.3.1. Bioethanol

Bamboo, rich in cellulose and hemicellulose, can undergo depolymerization into fermentable sugars under the action of cellulose hydrolase and hemicellulose hydrolase. These sugars can then be fermented into bioethanol [48]. However, the cell wall of bamboo is hard and dense; in the cell wall, cellulose fibril interacts with intermolecular hydrogen bonds and is wrapped and sealed by a polymer matrix of hemicellulose and lignin, which limits the conversion of polysaccharides to ethanol [49]. Therefore, in order to better deconstruct plant cell walls and improve the accessibility of polysaccharides to enzymes, pretreatment of bamboo is essential [50]. Given the high lignin content in bamboo, many pretreatment methods focus on lignin removal to enhance carbohydrate availability. Currently, pretreatment methods that can effectively remove lignin include physical (steam blasting [51]), chemical (acids [52], bases [53], peroxides [54], and other organic agents [55]), and biological (white rot fungi [56]) treatments. Steam blasting is an economical and environmentally friendly pretreatment method that mainly consists in extracting hemicellulose and changing the characteristics of cellulose. Gao et al. used the response surface method to perform steam explosion treatment and optimized green liquid pretreatment of bamboo samples of different ages [57]. The findings revealed that when compared with older samples, 1-year-old bamboo demonstrated complete biomass enzymatic saccharification, achieving a hexose yield of 100.0% (% cellulose). This resulted in a bioethanol yield of 20.3% (% dry biomass), surpassing all previously reported bamboo crafting methods. In the study by Huang et al., the highest lignin removal rate of bamboo pretreated with modified alkaline hydrogen peroxide was 79.25%, and the conversion rates of glucan and xylan were 96.76% and 97.38%, 7.4 times and 11.4 times those of untreated bamboo, respectively [58]. In addition, the mixture of hydrogen peroxide and citric acid can produce peroxy-citric acid with strong oxidation function, which can also effectively promote the enzymatic hydrolysis of bamboo residue and the production of bioethanol. Meng et al. found that after pretreatment of bamboo with peroxy-citric acid, lignin (95.36%) and xylan (55.41%)

were effectively removed and the enzymatic hydrolysis glycosylation rate was eight–nine times higher than that of bamboo with citric acid pretreatment [59]. Biological pretreatment has the advantages of mild conditions and low energy consumption and can be combined with chemical pretreatment to improve the efficiency of bamboo saccharification. Chu et al. achieved effective removal of lignin and xylan from bamboo using a collaborative treatment with sodium hydroxide and white rot bacteria, and the production of bioethanol without the use of commercial hydrolase [60].

### 2.3.2. Biosynthesis Gas

Biomass gasification, a green technology using a gasifier (air, oxygen, steam) to convert biomass into syngas ($H_2$, $CH_4$, $CO$, $CO_2$), has attracted more and more attention [61]. As a typical agricultural and forestry biomass resource, bamboo is widely used in the production of biosynthesis gas. Chen et al. compared the gasification efficiency of raw bamboo, carbonized bamboo, and highly volatile bituminous coal and found that the cold gasification efficiency of carbonized bamboo was 88% higher than that of raw bamboo [62]. Among various gasification technologies, steam gasification is more expensive than air gasification. However, it yields a higher calorific value and reduces tar production. Zheng et al. investigated the gasification properties of bamboo and polyethylene, revealing that bamboo can achieve the highest-quality syngas at 700 °C with a calorific value of 6.22 MJ/$Nm^3$ in air and steam atmospheres, while polyethylene requires a higher temperature [63]. Kakati et al. studied the effects of the steam-to-biomass ratio on gas composition and performance parameters under air–steam conditions, and the results showed that when the steam-to-biomass ratio was 0.35, the maximum hydrogen production was 37.12% and the lowest calorific value was 5.94 MJ/$Nm^3$ [64].

### 2.3.3. Solid Biofuels

Bamboo biomass can be utilized as solid biofuel through direct combustion or after thermochemical treatment. Its heating value or energy content is generally described on two bases, namely, high heating value (HHV, or gross calorific value) and low heating value (LHV, or net calorific value) [65].

#### Direct Combustion

Direct combustion of bamboo pellets or briquettes is the most massive and conventional way to convert biomass fuels to energy [66]. It has been proved that the HHV of various bamboo species shows limited difference, with favorable values of 17.67–21.00 kJ/g [12,67], which surpass those of most agricultural residues, straws, and grasses [68], such as corn cob (16.90 kJ/g), barley straw (16.81 kJ/g), rice straw (~15.50–16.81 kJ/g) [69,70], wheat straws (14.86–20.3 kJ/g) [71–74], grass (18,500–19,000 J/g) [75], etc. This would be extremely preferable for its commercial application as biofuels. Furthermore, biomass combustion, when monitored and controlled effectively, has the potential to curtail greenhouse gas emission and air pollution [76]. Lower nitrogen (N) and sulfur (S) contents are confirmed in bamboo residues compared with coal, sludge, and some other biomass types, leading to lower emission of NOx and SOx during combustion [67].

The combustion properties are greatly correlated with the moisture content and elemental composition of the feedstock. Liu et al. revealed that increased moisture content may not have a significant effect on the HHV of bamboo pellets but could facilitate the combustion rates and heat release rate by shortening the dwell time of bamboo particles and broadening the air gaps among particles [77]. They also pointed out that larger particle size with sufficient air circulation and oxygen also leads to the increment in combustion and heat release rates [78].

The element composition of bamboo varies with its species, positions, ages, etc. Therefore, these factors influence the combustion of bamboo correspondingly. The combustion characteristics of five bamboo species in India were compared, and a little variation in their HHV (18.7–19.6 kJ/g) was observed [79]. Further, it is reported that the combustion rate is

influenced by the position of bamboo residues in the order of shoot leaves > branches > bamboo leaves [67].

Torrefaction and Carbonization for Combustion

Considering that the energy density and fuel quality are generally low with direct combustion of biomass [80], thermochemical treatments before combustion, including torrefaction and carbonization (pyrolysis and hydrothermal carbonization), are advocated.

Torrefaction is a mild pyrolysis process that is generally performed at 200–300 °C, with limited or no oxygen [81]. It helps to achieve the energy densification and homogenization of biomass with decreased water uptake properties, no biomass decomposition, and reduced grinding energy requirements [82–86]. Further, hemicelluloses are largely decomposed from biomass during torrefaction [87], leading to improved HHV by lowering the moisture content and reducing the particle size [81]. The combustible properties of torrefied bamboo are connected with the composition of biomass and, most importantly, the temperature of torrefaction. Higher temperature results in the increment in HHV [81].

Pyrolysis, generally performed at temperatures of 300–1000 °C, is the most common method for the thermochemical treatment of bamboo. The feedstock can be converted into carbon-rich biofuels with optimized HHV, moisture content, and volatile contents; consequently, the combustion efficiency is greatly improved [88,89]. The as-obtained solid product is bamboo charcoal. The combustion features of bamboo are influenced by the pyrolysis conditions, namely, temperature and residence time, where the former one plays a more essential role than the latter one [90]. The vast majority of research confirms that higher pyrolysis temperature for bamboo biomass produces better combustion properties of the corresponding charcoal, with higher HHV, lower volatiles, and lower yield [66,90,91]. The volatile combustion stage of bamboo charcoal disappears at the temperature of ~350 °C [66]. However, the temperature of ~650 °C is also important, as the combustion process may shift to the high-temperature zone, leading to the decrease in both HHV and combustion reaction. By regulating the pyrolysis temperature, bamboo-derived charcoal can have a maximum HHV of 32.44 MJ/kg, which is comparable to that of high-grade coal, for example, medium-volatile bituminous (32.24 MJ/kg), and greatly exceeds that of original bamboo with direct combustion [66]. Although the residence time does not have an effect on the combustion properties of bamboo-derived charcoal comparable to that of pyrolysis temperature, the content of volatile matters, H/C, O/C, energy enrichment factor, HHV improvement, energy yield, and activation energy decrease, and the content of fixed carbon, carbon, ash, carbon densification factor, and fuel ratio (FR = Fixed Carbon $_{molded\ charcoal}$/Volatile Matter $_{molded\ charcoal}$) increase as the residence time increases [90].

However, pyrolysis is typically associated with significant energy consumption. To address this issue, hydrothermal carbonization, also known as wet torrefaction, is advocated as an alternative method [92–94]. Compared with untreated or torrefied biomass, the solid hydrochar produced through hydrothermal carbonization shows decreased O/C ratio, increased HHV, and improved hydrophobicity [95]. It is confirmed that hydrochar from bamboo exhibits better HHV and fixed carbon content than torrefied bamboo produced at the same temperature [80]. The temperature of hydrothermal carbonization is also important for hydrochar combustion, as the mass yield and volatile content decrease and HHV, fixed carbon content, and thermal degradation stability increase with the temperature.

Currently, bamboo-derived molded charcoal is the leading solid biofuel in China due to its high HHV. According to a report, ~0.26 million tons of bamboo charcoal was produced and burned in 2018 [77]; it was widely consumed in the fields of food heating, power plant heating, and metallurgy industry [96–98]. It is believed that bamboo will play an essential role in the market of solid biofuels continuously.

### 2.3.4. Bio-Oil

The liquid biofuel bio-oil is generally produced through the fast pyrolysis process at a high heating rate, over short residence time, at the temperature of 250–650 °C [99]. Solid char and gaseous biogas are obtained as by-products simultaneously during pyrolysis [100]. The yields of liquid, solid, and gas products by weight vary in the ranges of 30%–70%, 15%–50%, and 15%–20%, respectively [101,102]. The properties and yield of bio-oil are related to the reaction conditions and the feedstock [12,101,102]. It has been mentioned that bamboo has potential in bio-oil production, as it has the highest bio-oil yield among five kinds of biomass, including rice husk, rice straw, bamboo, sugarcane bagasse, and neem bark [99]. With the increment in pyrolysis temperature, the bio-oil yield increases as more char converts to bio-oil; however, a secondary cracking of bio-oil takes place, and more gaseous by-products can be obtained at higher temperatures [99]. The optimal reaction temperature for bio-oil production from bamboo sawdust is reported to be 405–450 °C, with a maximum yield of 70% [103]. The age of bamboo is also a noticeable factor in the yield and properties of bio-oil. Aging is good for increasing the yield, increasing the viscosity, and decreasing the pH of bio-oil. Further, HHV is also influenced by the age of bamboo; however, the effect varies with the species. Ding et al. found that the bio-oil produced from Pseudosasa amsbilis has HHV decreasing with age, while that from Pleiblastus chino has an increasing trend [104].

Generally, bio-oil products from bamboo contain acids, phenols, carbonylic compounds, furans, esters, ethers, and some other unidentified compounds, among which acid and phenols are the main components [105]. Due to the high content of oxygen, bio-oil shows lower HHV (16–19 MJ/kg) than crude oil (44 MJ/kg) [106]. Bio-oil also has low thermal instability, high moisture, and high corrosivity and viscosity [99,107]. Therefore, several methodologies are advocated to enhance the characteristics of bio-oil, including the utilization of microwave irradiation; the introduction of catalysts; and the combination of the two methods, namely, microwave-assisted catalytic fast pyrolysis. Further, co-pyrolysis and pretreatment are also beneficial for improving the quality and conversion efficiency of bio-oil [108,109].

Microwave irradiation can penetrate and then directly convert into heat throughout the entire feedstock particles [110]; therefore, heterogeneous reactions can be achieved, as the temperature distribution within the particles is more uniform than that in conventional production [111]. This undoubtfully leads to varied characteristics between microwave pyrolysis and conventional fast pyrolysis. It has been proved that microwave irradiation is favorable for bio-oil upgrading [112,113]. Recently, Giorcelli has explained the mechanism of the microwave-assisted pyrolysis of bamboo, which is correlated with the dehydration of cellulose, the degradation of hemicellulose, and the Maillard reaction in bamboo [114].

The introduction of catalysts, including zeolite catalysts, mineral catalysts, and carbon materials, is another approach for gathering high-quality bio-oil from bamboo [115–117]. Catalysts can remove oxygen from biomass with catalytic cracking reactions, thus benefitting the quality of bio-oil [118]. For example, HZSM-5 zeolites can increase the concentration of aromatic hydrocarbons with catalytic fast pyrolysis of bamboo sawdust [115]. Red mud is able to facilitate the deoxidation effect, reducing the acidity of bio-oil with a slight sacrifice of the yield of bamboo bio-oil [116].

Recently, microwave-assisted catalytic fast pyrolysis, which employs both microwave irradiation and catalysts, has gained growing attention [119]. Some catalysts are also good microwave absorbers, thus reducing the microwave output power [105]. Further, the above-mentioned co-pyrolysis with some plastics, such as tire and polypropylene, is also applied in microwave-assisted catalytic fast pyrolysis of bamboo for improving the yield and quality of bio-oil [120,121]. Plastics can reduce coke formation, increase the yield of bio-oil, and the proportion of aromatics [120]. However, till now, some more specific investigations still need to be carried out for better understanding the microwave-assisted catalytic fast pyrolysis process.

## 3. Indirect Application of Bamboo-Based Materials

### 3.1. Adsorption Materials

Due to the inherent porosity and unique structure of bamboo, utilizing waste generated during bamboo processing to create adsorbents has emerged as a novel approach to addressing issues of raw material wastage and environmental pollution. These adsorbents demonstrate excellent removal efficiency through various mechanisms, such as π-π interactions, hydrogen bonding, and electrostatic attraction. The term "pollutants" encompasses a wide range of substances that could have adverse effects on the environment, mainly including organic pollutants, inorganic pollutants, and gaseous pollutants. Bamboo-based adsorbents have shown remarkable performance in treating these pollutants, owing to the porous structure of bamboo materials, which enables efficient adsorption. This unique adsorption mechanism encompasses various interactions and offers the potential for sustainable solutions in environmental protection and pollution control, reducing the detrimental impact of harmful substances on the natural environment. In research on bamboo-based adsorbents, we will delve into these mechanisms to promote broader applications and further research advancements [13].

### 3.1.1. Organic Pollutants

In this section, we will discuss the removal of various organic pollutants using bamboo-based adsorbents. Organic pollutants encompass several substances that could potentially harm the environment, including organic dyes, pharmaceuticals, organic chemicals, and organic components found in various types of industrial wastewater [122–125].

For example, Li et al. utilized bamboo as the raw material, urea as the nitrogen source, and $KHCO_3$ as a green activator to successfully produce nitrogen-doped porous biochar with an in situ pyrolysis process. They examined the adsorption performance of this nitrogen-doped biochar using phenol and methylene blue as probe molecules. The results indicated that nitrogen doping effectively enhanced the adsorption capacity of biochar for phenol and methylene blue. Additionally, it was found that the biochar prepared under high-temperature conditions (973.15 K) with lower urea content exhibited the highest adsorption capacity for phenol and methylene blue, reaching 169.0 mg g$^{-1}$ and 499.3 mg g$^{-1}$, respectively [126]. The preparation of N-doped biochar and the mechanism are shown in Figure 4. Liu et al. used bamboo as the raw material and employed carbonization and KOH activation processes to create activated carbon with a high surface area, significant oxygen doping, and a three-dimensional layered porous structure. These distinctive characteristics resulted in its exceptional adsorption capacity for Rhodamine B [127]. Lv et al. used bamboo and acrylic acid as raw materials to perform one-pot hydrothermal carbonization with the assistance of ammonium persulfate, obtaining carboxylic acid-rich hydrochar. Subsequently, they activated hydrochar with a sodium hydroxide solution and conducted batch adsorption experiments to evaluate its adsorption capacity for methylene blue (MB). Despite its relatively small BET (Brunauer–Emmett–Teller) surface area, hydrochar exhibited outstanding adsorption performance owing to its abundant carboxylic acid groups [128]. In addition to organic dyes and chemicals, bamboo-based adsorbents also show excellent adsorption properties for drugs. For example, Guellati et al. employed the response surface methodology (RSM) with a central composite rotatable design (CCD) to optimize the preparation process of aluminum-dispersed bamboo activated carbon. They observed the percentage adsorption efficiency of ciprofloxacin hydrochloride (CIP) antibiotic and studied the response changes in relation to the activator ($AlCl_3$) concentration, activation temperature, and activation time. Under the conditions of 2.0 mol/L $AlCl_3$ concentration, 900 °C activation temperature, and 120 min activation time, the prepared adsorbent exhibited maximum CIP adsorption efficiency of 93.6 $\pm$ 0.36% (13.36 mg/g), indicating its excellent adsorption performance with ciprofloxacin hydrochloride [129].

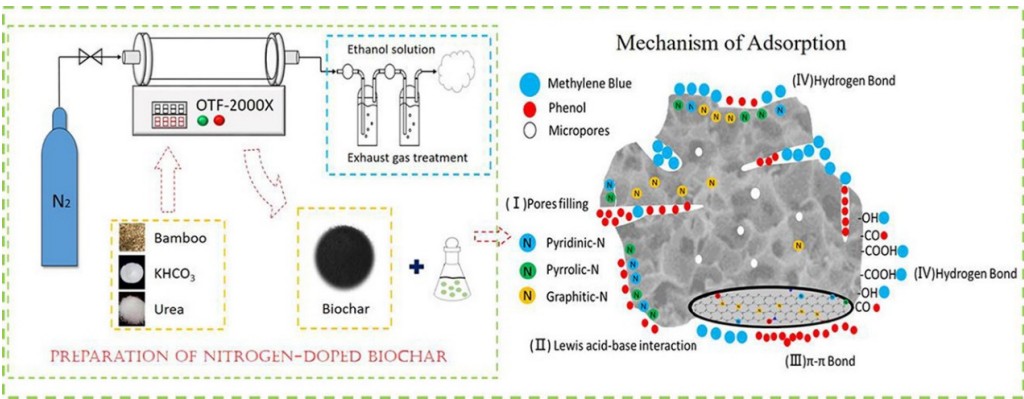

**Figure 4.** The preparation of N-doped biochar and mechanism of adsorption [126].

### 3.1.2. Inorganic Pollutants

Bamboo-based adsorbents exhibit excellent performance in effectively capturing and removing inorganic pollutants, including heavy-metal ions and inorganic fertilizers, owing to their porous structure and unique surface properties. Through various chemical reaction mechanisms, they reduce the concentration of inorganic pollutants in the environment, offering feasible solutions for environmental protection and water purification. For example, Hu et al. used bamboo shoot shells as the raw material to prepare pyrolyzed biochar at different calcination intensities. They investigated the reduction and immobilization of non-radioactive Cr (VI) as a chemical analog for radioactive Tc (VII). Batch adsorption experiments revealed that the pyrolyzed biochar exhibited strong adsorption capacity for Cr (VI), with maximum adsorption capacity of up to 63.11 mg/g. This suggests that bamboo-based biochar is a cost-effective and efficient material for Tc (VII) immobilization [130]. Tejada-Tovar et al. investigated the removal of Cd (II) ions from aqueous solutions using bamboo leaves as an adsorbent under physical conditions. They determined that bamboo leaves possess a rough, porous surface suitable for heavy-metal adsorption, with the adsorption process being facilitated by ion exchange. The optimal adsorption equilibrium was fitted to Langmuir and Freundlich models, with maximum removal capacity of 103.09 mg/g. The results demonstrate that bamboo leaves are effective in adsorbing Cd (II) [131].

In addition, bamboo-based biochar has been widely used to remove nitrate, ammonium, and phosphate ions produced by inorganic fertilizers in water. For example, Shao et al. used zirconia chloride octahydrate and cetyltrimethylammonium bromide (CTAB) to functionalize bamboo activated carbon (BAC), resulting in the preparation of Zr/CTAB/BAC. They observed that under conditions where the CTAB modifier concentration was 0.6 mM and the Zr/CTAB/BAC dosage ranged from 0.1 g to 1.0 g, the removal efficiency from simulated wastewater for phosphate ranged from 59.32% to 99.42%, and for nitrate, it ranged from 34.08% to 93.70%. Using NaOH as a regeneration agent, after four adsorption–desorption cycles, Zr/CTAB/BAC still maintained removal efficiency of over 70% for both phosphate and nitrate. Zr/CTAB/BAC exhibited noticeable capacity for removing phosphate and nitrate from water, with the potential to enhance the treatment of other anionic salts in wastewater, ultimately improving overall wastewater treatment efficiency [132]. Fan et al. delved into the adsorption characteristics of hydrated bamboo biochar for ammonium salts. Their findings underscore the pivotal role of solution pH in ammonia adsorption, with elevated ionic concentrations notably amplifying the adsorption capacity of bamboo biochar for ammonium salts. Furthermore, the generation of surface precipitates and complexes is conducive to the adsorption of ammonium by hydrated bamboo biochar. These outcomes substantiate that bamboo-based biochar serves as an efficacious adsorbent for eliminating ammonium from water [133].

### 3.1.3. Gaseous Pollutants

Bamboo is widely regarded as an exceptional material for combatting air pollution. Among the various environmental issues, carbon dioxide ($CO_2$) management stands out as one of the most intricate, given its implications in critical environmental problems, notably the greenhouse effect and climate change. Consequently, research into bamboo-based adsorbents predominantly centers on their $CO_2$ adsorption capacity, concurrently extending to the investigation of other pollutants, such as volatile organic compounds (VOCs), $NH_3$, NO, $H_2$, $CH_4$, $N_2$, and so on [13,134–136]. Research is primarily oriented toward identifying effective solutions to address the multifaceted challenges of the environment.

For example, Hou et al. introduced a sustainable material based on quaternized bamboo fibers and its application in direct air capture. Quaternized bamboo cellulose, after treatment, exhibited the ability to adsorb $CO_2$ under relative humidity conditions ranging from 60% to 80%, with quaternary ammonium adsorption efficiency exceeding 0.65. These results indicate that low-cost quaternized cellulose opens new possibilities for utilizing moisture-responsive $CO_2$ adsorbents in humid environments [137]. Ying et al. achieved the in situ synthesis of nitrogen-doped bamboo-based activated carbon (NBAC) by simply mixing bamboo charcoal (BC) with sodium amide powder and subsequently heating the mixture in a nitrogen atmosphere. They determined the $CO_2$ adsorption capacity of NBAC under conditions of 0 °C and 25 °C at a pressure of 1 bar, which reached ranges of 3.68 to 4.95 mmol/g and 2.49 to 3.52 mmol/g, respectively. Furthermore, through ten adsorption–desorption cycles, they confirmed the stable $CO_2$ adsorption performance of regenerated NBAC, demonstrating that this multifunctional NBAC exhibits excellent reproducibility and is an ideal candidate material for $CO_2$ capture and separation applications [138].

Regarding VOCs, Su et al. assessed the adsorption capacity of doped porous carbon materials (resin-based carbon and bamboo charcoal) for volatile organic compounds (VOCs) such as methanol and toluene with a combination of experimental investigations and theoretical calculations. The results revealed that the nitrogen-doped porous carbon materials possessed a significantly high specific surface area (2293.2 $m^2 \ g^{-1}$), exhibiting superior adsorption capacity for methanol (915.3 $mg \ g^{-1}$) and toluene (622.9 $mg \ g^{-1}$) compared with porous carbon materials derived from bamboo charcoal. This research suggests the potential of utilizing polarizable elements through doping for the adsorption of polar gases [139]. Gong et al. employed a novel, modified, and cost-effective adsorbent, bamboo charcoal (BC), for the removal of NO. The results revealed that various activation methods, such as $H_2O_2$ oxidation, alkali treatment, thermal modification, and metal loading, were advantageous for eliminating NO pollutants. The maximum NO removal efficiency using modified BC reached 85.9%, which was twice that achieved with the unmodified sample (38.1%). The increase in BET specific surface area and the presence of oxygen functional groups, particularly the augmentation of C=O and -COO groups, played a significant role in enhancing efficiency in NO removal. This study provides new insights into the understanding of BC adsorbents for controlling NO pollutants more effectively [135].

### 3.2. Electrode Materials (Supercapacitors)

Supercapacitors, also known as electrochemical capacitors, are of increasing scientific and public concern due to their high power density and energy density in batteries [140]. The electrode material is essential for supercapacitors' performance. Carbon materials are usually selected as electrode materials for supercapacitors due to their excellent conductivity, large specific surface area, and good stability [141].

Natural bamboo is a material with a hierarchical structure and vertically arranged pores. After carbonization treatment, it can be directly used as an electrode and is an ideal candidate for supercapacitor electrodes [142]. Li and Wu synthesized porous carbon derived from water bamboo for use as electrode material in supercapacitors, achieving a remarkable specific surface area of 2352 $m^2 \ g^{-1}$ [143]. Water bamboo was carbonized in $N_2$ atmosphere at 800 °C after KOH pretreatment. These water bamboo-derived porous carbon materials exhibited outstanding performance as supercapacitor materials, achieving

maximum specific capacitance of 268 F g$^{-1}$ at current density of 1 A g$^{-1}$ in a 6 M KOH electrolyte. Additionally, they demonstrated good capacity retention, reaching 97.28% over 5000 cycles at the current density of 10 A g$^{-1}$. This remarkable performance is attributed to their high specific surface area, pore volume, proper pore size distribution, and the presence of functionalized oxygen on the surface. However, the low specific energy of carbon-based supercapacitors is a major drawback that limits their application as supercapacitors. It is still challenging to increase specific energy and specific power simultaneously. Han et al. found that cross-cutting bamboo-derived carbon can markedly enhance specific capacitance, specific energy, and rate performance simultaneously. This improvement results from the cross-cutting of the long natural pore tunnels in bamboo and the enhancement in the hydrophilicity of the carbon surface [144]. The symmetric supercapacitor utilizing cross-cutting bamboo delivered high specific energy of 10.4 Wh kg$^{-1}$ in a 1 M Na$_2$SO$_4$ electrolyte within 0.47 s, at the relatively high specific power of 80,000 W kg$^{-1}$. Moreover, no capacitance loss was observed after 30,000 cycles at 50 A g$^{-1}$. Nguyen et al. created bamboo-derived hierarchical porous carbon (BHPC) under ambient air conditions using an eco-friendly, one-step, and easily scalable salt-templating strategy [145]. The obtained BHPC material exhibited a large specific surface area of 1296 m$^2$ g$^{-1}$ and a substantial total pore volume of 1.26 cm$^3$ g$^{-1}$, making it suitable for use as an electrode material in supercapacitors. The electrochemical evaluation of BHPC in a 6 M KOH electrolyte using a three-electrode system revealed excellent performance. It exhibited high specific capacitance of 394 F g$^{-1}$ at 1 A g$^{-1}$, along with robust rate capacity, showing 76.14% capacitance retention at 20 A g$^{-1}$. Furthermore, BHPC demonstrated outstanding durability, maintaining 81% capacitance retention over 10,000 cycles.

On the other hand, doping with heteroatoms, such as the introduction of transition metal compounds, can also significantly improve the conductivity and interfacial chemical properties of bamboo-derived carbon materials. Abbas et al. prepared a novel SiC/Pyrrolic-N doped carbon material synthesized from bamboo [146]. Using the inherent SiO$_2$ moieties in natural bamboo as a sacrificial template and taking advantage of the synergy of SiC and pyrrolic-N for faradaic redox reactions, the natural bamboo-based carbon materials displayed capacitance of 369 F g$^{-1}$ at 0.5 A g$^{-1}$ and 100% capacitance retention after 5000 cycles. Qiu et al. used a green activation strategy (CO$_2$-catalyzed induction of small doses of K$_2$CO$_3$) to prepare honeycomb layered porous carbon with excellent supercapacitor performance [147]. Chen et al. employed 2-methylimidazole (C$_4$H$_6$N$_2$) and sodium nitrate (NaNO$_3$) to convert bamboo powder to produce a N-, O-co-doped porous carbon material (BC-CNa) and used it as a zinc-ion hybrid supercapacitor (ZnHS) cathode material [15]. The synthesis of various bamboo-derived carbon materials and some characterizations are shown in Figure 5. Owing to the synergy effect, the ZnHS fabricated using BC-CNa as the cathode material demonstrated elevated specific capacity and energy density. BC-CNa based ZnHS achieved maximum specific capacity and energy density of 51.4 mA h g$^{-1}$ and 48.3 Wh kg$^{-1}$, respectively. Remarkably, the cycle stability is impressive, with 96% capacity retention having been observed after 90,000 cycles.

In general, owing to the unique multi-level and anisotropic three-dimensional pore structure advantages of bamboo, these low-cost and renewable bamboo-derived carbon materials possess remarkable conductivity, large specific surface area, and good stability. And their exceptional properties position them as promising candidates for building composite material pore structures for supercapacitors.

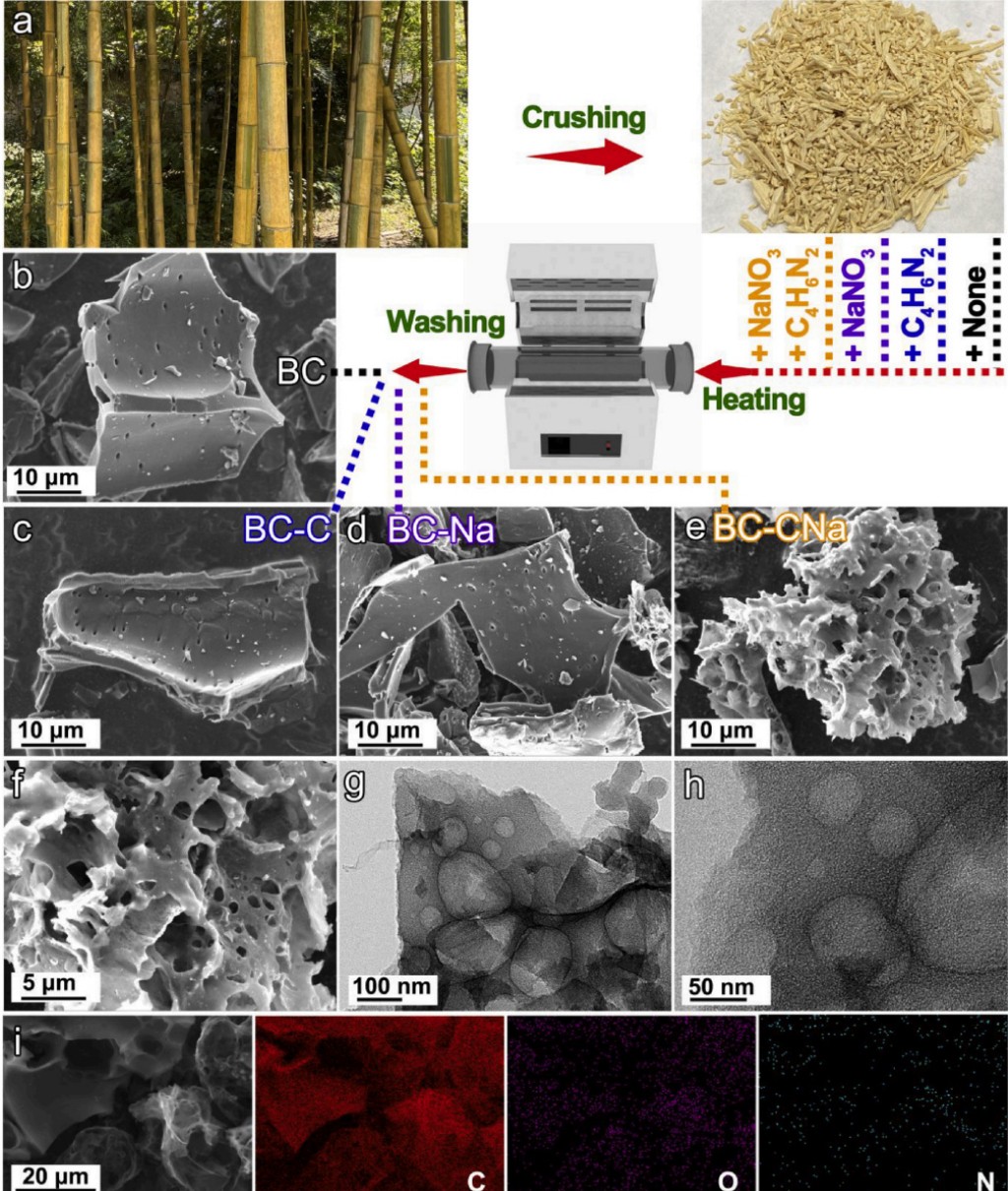

**Figure 5.** (**a**) Schematic illustrations of the synthesis of various bamboo-derived carbon materials. SEM images of (**b**) BC, (**c**) BC-C, (**d**) BC-Na, and (**e**,**f**) BC-CNa. (**g**,**h**) TEM images and (**i**) SEM elemental mapping of BC-CNa [15].

### 3.3. Electromagnetic-Shielding Materials

In light of the thriving electronics industry and the widespread utilization of personal computers and portable electronic devices, issues concerning electromagnetic interference and radiation are assuming an increasingly prominent role [148,149]. Considerable strides have been achieved in the exploration of diverse materials designed for electromagnetic interference shielding and attenuation over recent decades. These materials encompass carbon-based substances [150], conductive polymers [151], metals, as well as their respective oxides [152,153], showcasing their remarkable potential within the realm of electromagnetic interference mitigation. In recent years, as concerns regarding resource depletion have surfaced and environmental consciousness has gained momentum, bamboo resources have emerged as a subject of extensive interest [154]. These resources exhibit several remarkable features, such as cost effectiveness, sustainability, lightweight properties, a porous layered structure, and abundant availability. Bamboo serves not only as a

construction material for interior design and building projects but also as a functionally modified material with specific attributes achieved through composite enhancements. In comparison to other composite materials, bamboo-based composites not only preserve their inherent advantages but also introduce additional performance characteristics. Hence, bamboo-based materials for electromagnetic interference (EMI) shielding present a compelling and unpredictable landscape, given their distinctive attributes and the mounting demand for effective EMI-shielding solutions.

For example, Yan et al. employed waste bamboo materials as their study subject and prepared a series of two-dimensional biochar-based sheets with varying pore sizes through a combination of bamboo cellulose pyrolysis and subsequent chemical etching. The duration of etching directly influenced the pore size of the materials, subsequently impacting their dielectric properties. The results revealed that the biochar-based flakes obtained through a 6 h etching process (PGC-6) characterized by a macroporous structure exhibited excellent impedance matching and enhanced polarization effects, rendering them exceptional candidates as electromagnetic wave absorbers. PGC-6 displayed a minimum reflection loss (RL) value of 15.8 dB, with an absorption bandwidth extending up to 3.8 GHz at a thickness of 1.70 mm. This research presents a viable avenue for the resourceful utilization of agricultural and forestry waste materials [155]. Sun et al., using their advanced bamboo splitting technology, succeeded in obtaining bamboo pieces with uniformly distributed pore size. They then produced a large-scale honeycomb-like carbon tubular array (CTA) structure with controllable pore size, controllable graphitization level, and selectable electrical conductivity. Both simulation and experimental results indicated that the electromagnetic-shielding performance of CTAs is highly sensitive to the microchannel pore size and the angle at which electromagnetic energy is incident. This sensitivity arises from the differing propagation rates of induced electrons in different directions. Among the candidate materials, CTA-middle-1500 exhibited the best shielding performance for incident electromagnetic energy. The average SE/rho values in the vertical and parallel directions were 123.7 and 144.5 dB cm$^3$ g$^{-1}$, respectively, demonstrating its potential application as a lightweight, high-efficiency electromagnetic-shielding material [156]. The EM-shielding mechanism of CTAs is shown in Figure 6. Cai et al. utilized bamboo-derived lignin/cellulose mixtures with varying mass ratios as raw materials. Through a single-step pyrolysis process and surface modification of cellulose-derived graphene-like sheets using lignin-derived carbon particles, a series of heterogeneous structured carbon materials were prepared. BC-8, in particular, demonstrated remarkable electromagnetic wave (EMW) absorption capacity, featuring an impressive RLmin value of 49.4 dB and an effective absorption range spanning 14.4 GHz. When this particular coating was applied to the smooth surface of bamboo, the decorative finish achieved an outstanding SET value of 30.1 dB, suggesting the ability to attenuate more than 90% of electromagnetic energy. Furthermore, the surface exhibited enhanced hydrophobicity while preserving the original mechanical properties, suggesting its prospective application in electromagnetic attenuation within biomass-based home decoration [157]. Similarly, their team, under controlled conditions, employed p-toluenesulfonic acid hydrolysis and a homogenization process to obtain a wood cellulose-to-lignin mass ratio of 10:1. Subsequently, they subjected this wood cellulose to high-temperature pyrolysis at 1500 °C, yielding a carbon heterostructure with both fibrous and sheet-like morphologies, referred to as CH-10. The detailed findings highlight that the fibrous carbon structure, resembling fibers, exhibited high crystallinity and low defect density, mainly arising from the carbonization of cellulose within the lignocellulose (LC) nanofibers. On the other hand, graphite-like carbon sheets with higher defect density and lower crystallinity resulted from the carbonization of lignin within LC. Impressively, CH-10 demonstrated outstanding EMW absorption performance, characterized by a minimum RL value of −50.05 dB and an extensive absorption bandwidth of 4.16 GHz, achieved with a remarkably thin coating thickness of only 1.3 mm, making it an ideal candidate for electromagnetic wave (EMW) absorption [158]. Furthermore, Lou et al. employed bamboo as a model system, regulating its chemical composition using delignification, followed by in situ

modification with inorganic materials ($Fe_3O_4$) and high-temperature pyrolysis to prepare a series of magnetic biochar materials. These materials exhibited excellent electromagnetic wave (EMW) absorption capacity when their thickness was less than 2.00 mm, with the lowest RLmin value reaching −45.60 dB and a maximum absorption bandwidth (Fe) of 5.5 GHz. Additionally, one of the materials, designated as A4, demonstrated outstanding thermal stability, maintaining its EMW absorption performance even at high temperatures of up to 85 °C [159]. In another aspect of their research, bamboo-derived lignin was used for in situ pyrolytic modification of $Fe_3O_4$, resulting in the synthesis of monodisperse core–shell nanocrystals composed of a magnetic iron-based core and a graphitic carbon shell. The findings showed that the finely tuned Fe-C nanocrystal with a size below 13 nm exhibited an exceptionally broad electromagnetic (EM) absorption spectrum ranging from 8.4 to 18.0 GHz, with the RLmin value plunging as far as −47.11 dB at 8.0 GHz. This novel method for synthesizing sub-13 nm magnetic nanocrystals holds significant implications for the synthesis and design of nanocrystals and provides a new direction for research on high-performance electromagnetic absorbers [160].

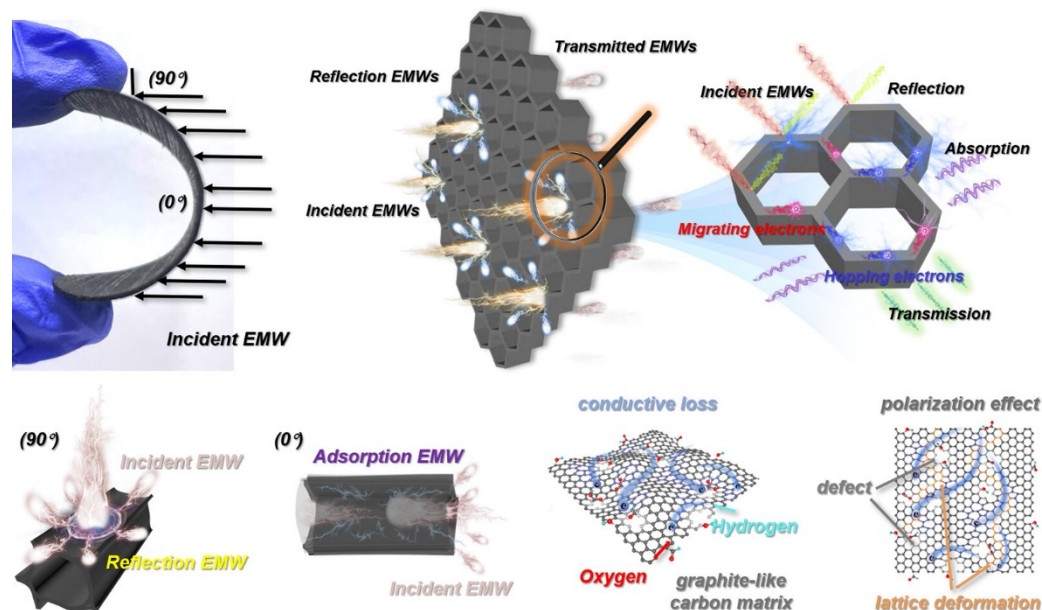

**Figure 6.** Schematic illustration of the EM-shielding mechanism of CTAs [156].

## 4. Future Prospectives

Relevant studies, including methods and properties, are listed in Table 1. As a future research direction, researchers should conduct in-depth research on the existing limitations of bamboo-based materials, thus focusing on the following: (1) maximizing recovery and enhancing the quality and uniformity of bamboo elements (such as radially sliced strips, veneers, broomed mats, and flattened culms); (2) facilitating the transition to the high-volume automated processing of elements and composites, further developing bamboo–wood hybrid composites for better bonding performance, and reducing overall product density; (3) developing standards for the bond qualification testing of bamboo–wood composites, adapted from those used for wood composites. In addition, bamboo-based materials have already proven to have tremendous potential in multiple domains, and further development of new bamboo-based materials could expand their applications into areas such as new battery technologies, catalytic materials, smart materials, and high-strength materials. This perspective offers deeper insights into the future of bamboo-based materials. (1) With the increasing demand for renewable energy and sustainable energy storage solutions, bamboo-based materials are poised to play a significant role in emerging battery technologies. The natural porous structure of bamboo fibers can be utilized to manufacture electrodes for lithium-ion batteries and supercapacitors, enhancing energy density and

cycle life while reducing dependence on rare metals, thus promoting sustainable battery technologies. (2) Bamboo-based materials can also serve as carriers for catalytic materials. Their rich surface functional groups and porosity make them an ideal choice for catalytic applications. Bamboo-based materials can find applications in organic synthesis, water treatment, and waste conversion, contributing to green chemistry and environmental protection. (3) The multifunctional properties of bamboo-based materials make them suitable for smart materials. For instance, bamboo fiber can be used to produce self-healing composite materials for applications in construction, aerospace, and automotive engineering. Furthermore, bamboo-based materials can be employed to develop smart sensing materials capable of monitoring structural changes and providing real-time data. (4) Bamboo fibers, renowned for their exceptional strength and lightweight characteristics, could revolutionize the construction and engineering sectors. Bamboo-based materials could replace traditional building materials, reducing energy consumption and carbon emissions. Moreover, these materials exhibit remarkable impact resistance and durability, making them suitable for manufacturing high-strength composite materials like carbon fiber composites.

**Table 1.** Existing applications of bamboo-based materials, and their preparation methods and properties.

| Application | Raw Materials | Methods | Performance | Ref. |
|---|---|---|---|---|
| Building materials | Raw bamboo | In situ lignin bonding | Shear strength: $\sim$4.4 MPa<br>Tensile strength: $\sim$300 MPa | [23] |
| | Cross-laminated bamboo | Planed and bonded with polyurethane adhesive | Compressive strength: 38.9 MPa,<br>Compressive modulus: 8056.6 MPa | [29] |
| | Bamboo stems | Delignification followed by water-assisted air drying | Tensile strength: $1.90 \pm 0.32$ GPa<br>Young's modulus: $91.3 \pm 29.7$ GPa<br>Toughness: $25.4 \pm 4.5$ MJ m$^{-3}$ | [37] |
| Furniture | Bamboo fiber | Hot pressing | Modulus: $9.66 \pm 1.14$ GPa<br>Improved fire and water resistance | [41] |
| | Bamboo powders | Self-bonding technology | Thickness swelling: 12.8%<br>Internal bonding strength: 0.71 MPa<br>Exhibited good paint film adhesion | [3] |
| | Moso bamboo | Non-notched-flattening technology | Cell wall modulus of elasticity: 20.1 GPa<br>Hardness: 0.89 GPa | [2] |
| Biofuel/-energy | Bambusa balcooa | Subjected to SO$_2$-impregnated steam pretreatment prior to enzymatic hydrolysis | Yield: 292 L of ethanol per dry ton of giant bamboo | [51] |
| | Neosino calamus affinis | Alkaline liquid hot water treatment | Yield: 30.9 g/100 g reducing sugars yielded;<br>9.6 g ethanol produced from 100 g of bamboo | [53] |
| | Moso bamboo | Dry torrefaction and hydrothermal carbonization | Calorific value: 28.29 MJ/kg<br>Energy yield: 59.77%<br>Fixed carbon content: 63.08% | [80] |
| Adsorption materials | Bamboo | In situ pyrolysis with KHCO$_3$ | Adsorption capacity for phenol: 169.0 mg g$^{-1}$<br>Adsorption capacity for MB: 499.3 mg g$^{-1}$ | [126] |
| | Bamboo pieces | Activation with AlCl$_3$ and (KOH/K$_2$CO$_3$) solution followed by carbonization | CIP adsorption efficiency: 13.36 mg/g | [129] |
| | Bamboo shoot shells | Torrefied at different intensities | Adsorption capacity for Cr (VI): 63.11 mg/g | [130] |

**Table 1.** *Cont.*

| Application | Raw Materials | Methods | Performance | Ref. |
|---|---|---|---|---|
| Electrode materials | Water bamboo | Carbonized in $N_2$ atmosphere at 800 °C after KOH pretreatment | Specific capacitance: 268 F $g^{-1}$ at a current density of 1 A $g^{-1}$ in 6 M KOH electrolyte. Capacity retention: 97.28% over 5000 cycles at current density of 10 A $g^{-1}$ | [143] |
| | Bamboo powder | Combustion between 2-methylimidazole and sodium nitrate | Specific capacity: 51.4 mA h $g^{-1}$. Energy density: 48.3 Wh $kg^{-1}$. Cycle stability: 96% capacity retention after 90,000 cycles | [15] |
| | Bamboo flakes | Carbonized and KOH-treated | Specific energy: 10.4 Wh $kg^{-1}$. Cycle stability: no capacitance loss after 30,000 cycles at 50 A $g^{-1}$ | [144] |
| Electromagnetic-shielding materials | Bamboo pulp | Pyrolysis followed by chemical etching | Minimum reflection loss value: 15.8 dB. Absorption bandwidth: 3.8 GHz | [155] |
| | Bamboo slices | Bamboo transverse splitting technology and carbonized | Average SE/rho values in the vertical direction: 123.7 dB $cm^3$ $g^{-1}$. Average SE/rho values in the parallel direction: 144.5 dB $cm^3$ $g^{-1}$ | [156] |
| | Bamboo-derived lignin | In situ pyrolytic modification of $Fe_3O_4$ | EM absorption spectrum: 8.4 to 18.0 GHz. Minimum RL value: 47.11 dB | [160] |

## 5. Conclusions

In summary, bamboo-based materials have promising prospects in various fields, including new battery technologies, furniture materials, building materials, and high-strength materials. Their sustainability, multifunctionality, and adaptability are poised to revolutionize materials science and engineering, contributing to sustainable development and environmental protection. Continued research and innovation are expected to unlock the full potential of bamboo-based materials, enabling them to play a more significant role in future technological advancements. Finally, bamboo's rapid growth, strength, versatility, and environmental benefits have made it an invaluable resource for sustainable living and responsible resource management. Its widespread use continues to expand, showcasing its extraordinary potential for addressing contemporary challenges and contributing to a greener and more sustainable future.

**Author Contributions:** Conceptualization, Z.Z. and Z.L.; writing—original draft preparation, Z.Z., N.Y., X.J., X.Z., R.X. and S.C.; writing—review and editing, Z.Z., C.L., L.X. and Z.L.; supervision, project administration, L.X. and Z.L. All authors have read and agreed to the published version of the manuscript.

**Funding:** This research was funded by National Key R&D Program of China (2023YFE0108300), Natural Science Foundation of Jiangsu Province (BK20221336), Jiangsu Agricultural Science and Technology Innovation Fund (CX(21)1010 and CX(23)3060), Jiangxi Forestry Bureau Forestry Science and Technology Innovation Special Project (No. 202240), and Open funding from the Key Laboratory for Protected Agricultural Engineering in the Middle and Lower Reaches of Yangtze River [ZX(23)3104].

**Data Availability Statement:** No new data were created.

**Conflicts of Interest:** The authors declare no conflict of interest.

## Abbreviations

| | Abbreviations | | Abbreviations |
|---|---|---|---|
| Fiber-reinforced polymer | FRP | Central composite rotatable design | CCD |
| Volatile organic compounds | VOCs | Ciprofloxacin hydrochloride | CIP |
| Bamboo self-bonded composites | BSCs | Cetyltrimethylammonium bromide | CTAB |
| Scanning electron microscope | SEM | Bamboo activated carbon | BAC |
| Theory of Inventive Problem Solving | TRIZ | Nitrogen-doped bamboo-based activated carbon | NBAC |
| High heating value | HHV | Bamboo charcoal | BC |
| Low heating value | LHV | Bamboo-derived hierarchical porous carbon | BHPC |
| Fuel ratio | FR | Electromagnetic interference | EMI |
| Methylene blue | MB | Reflection loss | RL |
| Brunauer–Emmett–Teller | BET | Carbon tubular array | CTA |
| Response surface methodology | RSM | Electromagnetic wave | EMW |

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
