# Peer review of "Modification and Application of Bamboo-Based Materials: A Review—Part II: Application of Bamboo-Based Materials"

_forests, doi:10.3390/f14112266_

Round 1
Reviewer 1 Report
Comments and Suggestions for Authors
comment in the attachment

Author Response
Respond to Reviewer #1
Modification and application of bamboo-based materials: A review—Part II: Application of bamboo-based materials
Due to the continuously increasing demand for building materials across the world, it is necessary to use renewable materials in place of the existing nonrenewable materials in construction projects. Bamboo is a fast-growing flowering plant that may be used as a renewable material in construction. The use of bamboo in the construction of buildings can improve its long-term carbon fixation capacity and economic benefits. Although bamboo has the advantages of superior performance, low carbon content, high energy-saving and emission-reducing capacity, bamboo is an anisotropic material, which has many factors affecting its material performance, large variability of material performance, lack of systematic research, and the use of bamboo as the main building material is not always limited.
Publications on bamboo composites have increased significantly over the past two decades, with a concurrent increase in the sub-topic of bamboo bonding. The strength and durability of bamboo bonds and bamboo-wood bonds play a critical role in the performance of the resulting composite products. The natural variations in bamboo permeability, density, and chemical composition create unique challenges for bamboo bonding with adhesives, which are designed for wood products. Compared to wood adhesion, bamboo bonding is generally much more difficult and poorly understood.
- Compared with wood, bamboo is high in density and high in starch, wax and silica content. In addition, chemical and microstructural gradients across the bamboo culm wall create challenges for bonding with adhesives formulated for wood. How was this problem solved?
Response: Thank you for your comments. To address this challenge, researchers employed various innovative approaches. Firstly, they developed adhesives tailored to the specific characteristics of bamboo, ensuring stronger bonding in the presence of chemical and microstructural gradients across the bamboo culm wall. Secondly, by optimizing the bonding process and adjusting the temperature and pressure of the adhesives, they ensured uniform and durable bonding on the uneven surface of bamboo. These improvements in processes enabled more effective utilization of bamboo in diverse applications, including construction and furniture manufacturing. This comprehensive approach not only addressed the distinct chemical and structural features of bamboo compared to wood but also propelled the widespread use of bamboo as a sustainable building material. We listed two articles detailing the research of bamboo adhesives and their bonding process, as follows:
- Wang, J.H.; Wang, R.S.; Ji, X.X.; Jin, C.D.; Yan, Y.T. Enhancing and toughening bamboo interfacial bonding strength by reactive hyperbranched polyethyleneimine modified phenol formaldehyde resin adhesive. JMR&T. 2023, 26, 8213-8228.
- Dong, W.Q.; Wang, Z.Q.; Chen, G.J.; Wang, Y.; Huang, Q.Z.; Gong, M. Bonding performance of cross-laminated timber-bamboo composites. J. Build. Eng. 2023, 63, 105526.
- That building code approval and application of bamboo and bamboo-wood composites in wood-frame construction is currently limited by the lack of agreed standards and protocols for accurate bond durability assessment. What are the research directions on this topic?
Response: In addressing the current limitations related to building code approval and application of bamboo and bamboo-wood composites in wood-frame construction, researchers may focus on several key research directions:
- Standardization and Protocol Development:
-Develop standardized testing protocols for accurately assessing the bond durability of bamboo and bamboo-wood composites.
-Establish industry-wide standards to ensure consistency and reliability in testing methods.
- Durability Studies:
-Conduct comprehensive durability studies to evaluate the long-term performance of bamboo and bamboo-wood composites in various environmental conditions.
-Investigate the effects of factors such as moisture, temperature, and exposure to different climates on bond strength and structural integrity.
- Material Characterization:
-Enhance material characterization techniques to better understand the chemical and physical properties of bamboo and bamboo-wood composites.
-Explore advanced imaging and analysis methods for a more detailed assessment of bond interfaces.
- Adhesive Formulation and Optimization:
-Research and develop adhesives tailored to the unique properties of bamboo.
-Optimize adhesive formulations for improved bonding performance, considering factors like flexibility, strength, and resistance to environmental factors.
- Structural Design and Modeling:
-Develop structural design guidelines specifically tailored for bamboo and bamboo-wood composite materials.
-Utilize advanced modeling techniques, such as finite element analysis, to simulate and predict the behavior of these materials in different structural configurations.
- Lifecycle Assessments:
-Conduct lifecycle assessments to evaluate the environmental impact of using bamboo and bamboo-wood composites in wood-frame construction.
-Compare the sustainability of these materials with traditional wood and other construction materials.
- Collaboration and Knowledge Exchange:
-Encourage collaboration among researchers, industry professionals, and regulatory bodies to facilitate knowledge exchange and the development of best practices.
-Establish forums for sharing research findings, case studies, and successful applications of bamboo in construction.
By addressing these research directions, the field can advance toward establishing robust standards, improving material performance, and expanding the application of bamboo and bamboo-wood composites in wood-frame construction. I sincerely hope that my answer has satisfied your questions.
I propose, focus areas for future bamboo composite research should be: 1) maximize recovery and enhance quality and uniformity of bamboo elements (such as radially sliced strips, veneers, broomed mats and flattened culms), 2) facilitate the transition to high volume, automated processing of elements and composites, similar in manner to engineered wood products, further develop bamboo-wood hybrid composites for better bonding performance, customization of flexural moduli (MOE and MOR), and reduction in overall product density, and 3) develop standards for bond qualification testing of bamboo-wood composites, adapted from those used for wood composites
Response: Thank you very much for your comments, your suggestions are very professional and comprehensive, we fully agree with your views and have added relevant descriptions in the article.

Reviewer 2 Report
Comments and Suggestions for Authors
Dear Authors,
in my opinion, the manuscript you have prepared is comprehensive and valuable due to the broad characterization of bamboo applications. Thus, in my opinion, the manuscript is worth to be published. Just 3 remarks, that have to be taken into account before publication, please find below:
- line 229 – „Suhaily et al. [47] Laminated bamboo hybrid with oil palm veneer…” – unclear
- line 233 – the „percentage” word is unnecessary since even if you measure the water absorption in g/g, the feature characteristics and influence on others will be the same
- lines 237-238 – please be more precise and add information on what composite you mean. Even if it was mentioned above, please add this info here when characterizing this material.
Best regards!
Author Response
Respond to Reviewer #2
Dear Authors,
In my opinion, the manuscript you have prepared is comprehensive and valuable due to the broad characterization of bamboo applications. Thus, in my opinion, the manuscript is worth to be published. Just 3 remarks, that have to be taken into account before publication, please find below:
- line 229 – “Suhaily et al. [47] Laminated bamboo hybrid with oil palm veneer…” – unclear
Response: Thank you for your comments, we have changed it for a clearer description, as follows: “Suhaily et al. analyzed the physical, mechanical and morphological properties of a five-layer composite material composed of dry bamboo strips and oil palm trunk veneer in order to explore its feasibility as an alternative material for furniture manufacturing.”
- line 233 – the “percentage” word is unnecessary since even if you measure the water absorption in g/g, the feature characteristics and influence on others will be the same
Response: Thank you very much for your suggestion, we have replaced it with “behavior”.
- lines 237-238 – please be more precise and add information on what composite you mean. Even if it was mentioned above, please add this info here when characterizing this material.
Response: We appreciate your comments very much, we have added the specific materials, and the content is revised as follows: “The morphology of the composite which was composed of dry bamboo strips and oil palm trunk veneer verified the high content of silica in oil palm trunk veneer.”
Best regards!

Reviewer 3 Report
Comments and Suggestions for Authors
The manuscript reviews the application of bamboo-based materials. It is a very exhaustive study, very well organized and written.
It can be published after operating minor corrections, according to the following suggestions:
- lines 120-121: part of the sentence is not clear ”The embedding of steel bars makes the bamboo in the compression zone of the slabs and beams develop with higher efficiency”
- lines 176-177: ”In terms of theory, Fu et al. [43] studied finite element analysis method the furniture in the condition of load can be analyzed.”-not clear
- lines 219-221: the sentences should be merged ”In terms of design, in order to ensure greater stability of the bamboo reinforcements and further to increase the durability of the benches of the furniture. Vengala et al. [46] studied the design of the benches based on ergonomics to maintain a correct posture and to increase the comfort level of potential users.”
- line 241: ”From this research, believe that the continuous efforts....”- a word is missing
-lines 246-247: ”Recombinant bamboo itself not only is better than ordinary timber in mechanical strength, grain color is beautiful, and its process ability is similar to hardwood [49].”- use but for the second part of the sentence
- line 271: ”The highest of all previously reported bamboo crafts.” has no meaning
-lines 308-311: it is not necessary to define the LHV and HHV, in my opinion
- line 369: what means fuel ratio? Maybe it is air-fuel ratio.
- lines 371-372: ”However, pyrolysis is usually accompanied with high energy consumption and hydrothermal carbonization, also refer as wet torrefaction, is thereafter promoted [94-96].”-not clear
- line 469: ”The preparation of N-doped biochar and mechanism shown in Figure 4.” - a verb is missing
-line 478: what means BET surface are? It should be explained.
-lines 600-602: ”And obtained BHPC material has large specific surface area (1296 m2 g−1) and large total pore volume (1.26 cm3 g−1) which make it applicable for supercapacitor as an electrode material” -rephrase the sentence
Comments on the Quality of English LanguageEnglish is good, but there are some sentences that need to be revised.
Author Response
Respond to Reviewer #3
The manuscript reviews the application of bamboo-based materials. It is a very exhaustive study, very well organized and written.
It can be published after operating minor corrections, according to the following suggestions:
- lines 120-121: part of the sentence is not clear “The embedding of steel bars makes the bamboo in the compression zone of the slabs and beams develop with higher efficiency”
Response: Thanks for your suggestion, we have changed it to a clearer description as follows: “The ultimate bearing capacity, flexural rigidity, ductility and material utilization efficiency of bamboo beams can be effectively improved by embedding steel bars and prestressed steel bars…”
- lines 176-177: “In terms of theory, Fu et al. [43] studied finite element analysis method the furniture in the condition of load can be analyzed.”-not clear
Response: Thank you very much for your opinion. I'm sorry that this is caused by our negligence. We have revised it to the accurate content as follows: “In order to explore the application of bamboo in furniture, Fu et al. used ANSYS finite element analysis software to compare the stress and deformation of reconstituted bamboo under load.”
- lines 219-221: the sentences should be merged “In terms of design, in order to ensure greater stability of the bamboo reinforcements and further to increase the durability of the benches of the furniture. Vengala et al. [46] studied the design of the benches based on ergonomics to maintain a correct posture and to increase the comfort level of potential users.”
Response: Thank you for your suggestion, we have merged it into “In order to assess the feasibility of employing environmentally friendly composite materials for street furniture, Vengala et al. conducted pertinent research, ensuring greater stability of bamboo reinforcements and further enhancing the durability of furniture benches.”
- line 241: “From this research, believe that the continuous efforts....”- a word is missing
Response: Thanks for your suggestion, we have revised this sentence.
-lines 246-247: “Recombinant bamboo itself not only is better than ordinary timber in mechanical strength, grain color is beautiful, and its process ability is similar to hardwood [49].”- use but for the second part of the sentence
Response: We appreciate your advice and we have corrected this sentence.
- line 271: “The highest of all previously reported bamboo crafts.” has no meaning
Response: Thanks for your comments, we have made modifications as follows: “This resulted in a bioethanol yield of 20.3% (% dry biomass), surpassing all previously reported bamboo crafting methods.”
-lines 308-311: it is not necessary to define the LHV and HHV, in my opinion
Response: Thanks for your suggestion, we have deleted this part.
- line 369: what means fuel ratio? Maybe it is air-fuel ratio.
Response: Thank you for your comments, the fuel ratio (FR) calculation formula is as follow:
FR = Fixed Carbon molded charcoal/Volatile Matter molded charcoal
We have added this formula in the paper.
- lines 371-372: “However, pyrolysis is usually accompanied with high energy consumption and hydrothermal carbonization, also refer as wet torrefaction, is thereafter promoted [94-96].”-not clear
Response: Thank you very much for your suggestion, we have rephrased it into a clearer description as follows: “However, pyrolysis is typically associated with significant energy consumption. To address this issue, hydrothermal carbonization, also known as wet torrefaction, is advocated as an alternative method.”
- line 469: “The preparation of N-doped biochar and mechanism shown in Figure 4.” - a verb is missing
Response: Thank you very much for your careful review of our manuscript, we have made revisions.
-line 478: what means BET surface are? It should be explained.
Response: Thank you for your comments. The specific surface area refers to the total area possessed by a unit mass of material and is a commonly used characterization method in the field of materials science. The BET surface area specifically denotes the specific surface area obtained through the method established by the three scientists Brunauer, Emmett, and Teller, represented by their initials as BET. We have added the relevant description in the article.
-lines 600-602: “And obtained BHPC material has large specific surface area (1296 m2 g−1) and large total pore volume (1.26 cm3 g−1) which make it applicable for supercapacitor as an electrode material” -rephrase the sentence
Response: Thanks for your suggestion, we have rephrased this sentence as “The obtained BHPC material exhibits a large specific surface area of 1296 m² g⁻¹ and a substantial total pore volume of 1.26 cm³ g⁻¹, making it suitable for use as an electrode material in supercapacitors.”.

Reviewer 4 Report
Comments and Suggestions for Authors
1. Research derivatives can be listed out appropriately
2. Include a table to consolidate the direct and indirect applications and provide the potential research gap and expert suggestions.
3. Novelty stated that indirect application consolidation is the core so reduce section 2 and increase the section 3 contents or research outcomes.
4. Future prospects can be enlarged and can be converted into a separate section
5. English can be improved, in some instances like ("we not only explore" in abstract)
6. Abstract can be expanded.
7. Novelty can be included in the introduction section at the end
Comments on the Quality of English Language
Need English correction
Author Response
Respond to Reviewer #4
- Research derivatives can be listed out appropriately.
Response: Thanks for your comments, we have listed all abbreviations in the article for readers to better understand, the table is as follows:
|
|
Abbreviations |
|
Abbreviations |
|
Fiber-reinforced polymer |
FRP |
Central composite rotatable design |
CCD |
|
Volatile organic compounds |
VOC |
Ciprofloxacin hydrochloride |
CIP |
|
Bamboo self-bonded composites |
BSCs |
Cetyltrimethylammonium bromide |
CTAB |
|
Scanning electron microscope |
SEM |
Bamboo activated carbon |
BAC |
|
Theory of Inventive Problem Solving |
TRIZ |
Nitrogen-doped bamboo-based activated carbon |
NBAC |
|
High heating value |
HHV |
Bamboo charcoal |
BC |
|
Low heating value |
LHV |
Bamboo-derived hierarchical porous carbon |
BHPC |
|
Fuel ratio |
FR |
Electromagnetic interference |
EMI |
|
Methylene blue |
MB |
Reflection loss |
RL |
|
Brunauer-Emmett-Teller |
BET |
Carbon tubular array |
CTA |
|
Response surface methodology |
RSM |
Electromagnetic wave |
EMW |
- Include a table to consolidate the direct and indirect applications and provide the potential research gap and expert suggestions.
Response: We appreciate your suggestions and we have added a table at the end of the article containing various applications of bamboo-based materials. The potential research gaps and suggestions have also been added to the corresponding section of the article. The table is as follows:
|
Application |
Raw materials |
Methods |
Performance |
Ref. |
|
Building materials |
Raw bamboo |
In situ lignin bonding |
Shear strength: ∼4.4 MPa Tensile strength: ∼300 MPa |
[24] |
|
Cross-laminated bamboo |
Planed and bonded with polyurethane adhesive |
Compressive strength: 38.9 MPa, Compressive modulus: 8056.6 MPa |
[30] |
|
|
Bamboo stems |
Delignification followed by water-assisted air-drying. |
Tensile strength: 1.90±0.32 GPa Young's modulus: 91.3±29.7 GPa Toughness: 25.4±4.5 MJ m-3 |
[38] |
|
|
Furniture |
Bamboo fiber |
Hot pressing |
Modulus: 9.66±1.14GPa Improved fire and water resistance |
[42] |
|
Bamboo powders |
Self-bonding technology |
Thickness swelling :12.8% Internal bonding strength :0.71MPa. Exhibited good paint film adhesion |
[18] |
|
|
Moso bamboos |
Non-notched flattening technology |
Cell wall modulus of elasticity: 20.1 GPa Hardness: 0.89 GPa |
[44] |
|
|
Bio-fuel/energy |
Bambusa balcooa |
Subjected to SO2 impregnated steam pretreatment prior to enzymatic hydrolysis |
Yield: 292 L of ethanol per dry ton of giant bamboo |
[53] |
|
Neosino calamus affinis |
Alkaline liquid hot water treatment |
Yield: 30.9 g/100 g reducing sugars yielded; 9.6 g ethanol produced from 100 g bamboo |
[55] |
|
|
Moso bamboo |
Dry torrefaction and hydrothermal carbonization |
Calorific value: 28.29 MJ/kg, Energy yield: 59.77% Fixed carbon content: 63.08% |
[82] |
|
|
Adsorption materials |
Bamboo |
In-situ pyrolysis with KHCO3 |
Adsorption capacities for phenol: 169.0 mg g-1 Adsorption capacities for MB: 499.3 mg g-1 |
[128] |
|
Bamboo pieces |
Activation with AlCl3 and (KOH/K2CO3) solution followed by carbonization |
CIP adsorption efficiency: 13.36 mg/g |
[131] |
|
|
Bamboo shoot shells |
Torrefied at different intensities |
Adsorption capacity for Cr (VI): 63.11 mg/g |
[132] |
|
|
Electrode materials |
Water bamboo |
Carbonized under N2 atmosphere at 800 °C after KOH-pretreatment |
Specific capacitance: 268 F g−1 at a current density of 1 A g−1 in 6 M KOH electrolyte Capacity retention: 97.28% over 5000 cycles at a current density of 10 A g−1 |
[145] |
|
Bamboo powder |
Combustion between 2-methylimidazole and sodium nitrate |
Specific capacity: 51.4 mA h g–1 Energy density: 48.3 Wh kg–1 Cycle stability: 96 % capacity retention after 90,000 cycles |
[15] |
|
|
Bamboo flakes |
Carbonized and KOH-treated |
Specific energy: 10.4 Wh kg−1 Cycle stability: no capacitance loss after 30,000 cycles at 50 A g−1 |
[146] |
|
|
Electromagnetic Shielding materials |
Bamboo pulp |
Pyrolysis followed by chemical etching |
Minimum reflection loss value: 15.8 dB Absorption bandwidth: 3.8 GHz |
[157] |
|
Bamboo slices |
Bamboo transverse splitting technology and carbonized |
The average SE/rho values in the vertical directions: 123.7 dB cm³ g⁻¹ The average SE/rho values in the parallel directions: 144.5 dB cm³ g⁻¹ |
[158] |
|
|
Bamboo-derived lignin |
In-situ pyrolytic modification of Fe3O4 |
EM absorption spectrum: 8.4 to 18.0 GHz Minimum RL value: 47.11 dB |
[162] |
- Novelty stated that indirect application consolidation is the core so reduce section 2 and increase the section 3 contents or research outcomes.
Response: Thank you very much for your suggestions. We have deleted and refined the section 2 part and supplemented the section 3 part to make the structure of the article more reasonable.
- Future prospects can be enlarged and can be converted into a separate section
Response: Thanks very much for your suggestion, we have expanded the future prospects section and made it a separate heading.
- English can be improved, in some instances like ("we not only explore" in abstract)
Response: Thanks for your suggestion, we have checked and improved the English of the full text.
- Abstract can be expanded.
Response: Thanks for your comments, we have expanded the abstract. The expanded abstract is as follows: “Bamboo, with its inherently porous composition and exceptional renewability, stands as a symbolic embodiment of sustainability. The imperative to fortify the utilization of bamboo-based materials becomes paramount for future developmental. These materials not only find direct ap-plications in construction and furniture sectors but also exhibit versatility in burgeoning domains such as adsorption materials and electrode components, thereby expanding their consequential influence. This comprehensive review meticulously delves into both their explicit applications and the nuanced panorama of derived uses, thereby illuminating the multifaceted nature of bamboo-based materials. Beyond their current roles, these materials hold promise in addressing environmental challenges and serving as eco-friendly alternatives across diverse industries. Lastly, we provide some insights into the future prospects of bamboo-based materials, which are poised to lead the way in further development. In conclusion, bamboo-based materials hold immense potential across diverse domains and are set to play an increasingly pivotal role in sustainable development.”
- Novelty can be included in the introduction section at the end
Response: Thanks for your suggestion, we have added the novelty description in the end of introduction section.
Round 2
Reviewer 1 Report
Comments and Suggestions for Authors